# Atomic-Resolution Structures and Mode of Action of Clinically Relevant Antimicrobial Peptides

**DOI:** 10.3390/ijms23094558

**Published:** 2022-04-20

**Authors:** Surajit Bhattacharjya, Sk Abdul Mohid, Anirban Bhunia

**Affiliations:** 1School of Biological Sciences, 60 Nanyang Drive, Nanyang Technological University, Singapore 637551, Singapore; 2Department of Biophysics, Bose Institute, Unified Academic Campus, Saltlake, Sector V, EN 80, Kolkata 700091, India; skabdulmohid1992@gmail.com (S.A.M.); anirbanbhunia@gmail.com (A.B.)

**Keywords:** antibiotics, multidrug resistant (MDR) bacteria, MDR Gram negative bacteria, antimicrobial peptides (AMPs), lipopolysaccharide (LPS), mechanism of AMPs

## Abstract

Global rise of infections and deaths caused by drug-resistant bacterial pathogens are among the unmet medical needs. In an age of drying pipeline of novel antibiotics to treat bacterial infections, antimicrobial peptides (AMPs) are proven to be valid therapeutics modalities. Direct in vivo applications of many AMPs could be challenging; however, works are demonstrating encouraging results for some of them. In this review article, we discussed 3-D structures of potent AMPs e.g., polymyxin, thanatin, MSI, protegrin, OMPTA in complex with bacterial targets and their mode of actions. Studies on human peptide LL37 and de novo-designed peptides are also discussed. We have focused on AMPs which are effective against drug-resistant Gram-negative bacteria. Since treatment options for the infections caused by super bugs of Gram-negative bacteria are now extremely limited. We also summarize some of the pertinent challenges in the field of clinical trials of AMPs.

## 1. Introduction

At present, antibiotics are the major drugs administered to eliminate infectious diseases caused by bacteria and other microbes. However, our dependence on the frontline antibiotics has now been challenged by the steady rise of bacterial antimicrobial resistance (AMR) problems. Although, the emergence of antibiotic resistance pathogens is a natural phenomenon [1,2,3]. Penicillinase, a bacterial enzyme that hydrolyzes penicillin antibiotic, was identified even before the therapeutic approval of penicillin [1,2,3]. Notably, resistant bacterial strains could be isolated within few years of the introduction of an antibiotic, for example the first methicillin resistant *S. aureus* (MRSA) was detected in 1961 only two years after methicillin was introduced [1,2,3]. Overuse and misuse of antibiotics over the decades have now been escalated to the occurrence of drug resistant, multi-drug-resistant (MDR) and extremely drug-resistant (XDR) bacterial pathogens. Bacterial AMR has been recognized as one of the major health issues of this century. Commissioned by the UK government, the O’Neill report made a thorough analysis of AMR and provided an estimate of 10 million deaths/year by 2050 [4]. In 2019, Centers for disease control and prevention (CDC) of USA indicated 2.8 million of antibiotic-resistant infections resulting in 35,000 annual deaths [5]. A very recent comprehensive analyses on the global burden of bacterial antimicrobial resistance in 2019 has reported 4.95 million deaths associated with bacterial AMR including 1.27 million more deaths attributable to bacterial AMR [6]. The six leading bacterial pathogens *Escherichia coli*, *Staphylococcus aureus*, *Klebsiella pneumoniae*, *Streptococcus pneumoniae*, *Acinetobacter baumannii*, and *Pseudomonas aeruginosa* have caused 929,000 deaths attributable to AMR and 3.57 million fatalities associated with AMR in 2019 [6]. In addition, AMR-associated deaths are found to be prevalent due to infections caused by six other pathogens namely *Mycobacterium tuberculosis*, *Enterococcus faecium*, *Enterobacter* spp., *Streptococcus agalactiae* or group B *Streptococcus*, *Salmonella* typhi, and *Enterococcus faecalis* [6]. Among bacterial AMR, infections caused by Gram-negative strains are the most difficult to be cured [7,8,9]. Indeed, the World Health Organization (WHO) emphasizes lack of potent drug candidates due to limited discovery of new antibiotics. Most antibiotics are simply modifications of existing drugs and are ineffective against resistant Gram-negative bacteria [10]. As a matter of fact, Gram-negative bacteria are intrinsically more impervious to several conventional antibiotics compared to Gram-positive bacteria. A trove of frontline antibiotics e.g., vancomycin, rifampicin, erythromycin, novobiocin etc., which can kill Gram-positive bacteria are not used for the treatment of Gram-negative infections [7,8,9,10,11,12,13]. Permeability barrier of the outer membrane LPS and porins can significantly restrict entry of large scaffold antibiotics to the bacterial targets [7,8,9,10,11,12,13]. Hence, there is an urgent need for the discovery and development of novel drugs which will be effective in the treatment of bacterial AMR particularly drug-resistant Gram-negative bacteria.

In the era of bacterial AMR and the drying pipelines of new small molecule antibiotics, antimicrobial peptides (AMPs) provide certain hopes to fight against MDR pathogens [14,15,16]. AMPs, also termed as host defense peptides (HDPs), are evolutionarily conserved arsenals of all organisms to fend microbial invasion [17,18,19]. These molecules constitute an integral component of innate immune system in higher organisms. AMPs are often synthesized as large precursor proteins and are released upon digestion caused by host proteases [17,18,19]. Most AMPs are bequeathed with an impressive broad spectrum of anti-pathogenic activity including Gram-negative and Gram-positive bacteria, fungi, viruses, and parasites [20,21,22]. In addition, activities akin to immunomodulatory, anti-inflammatory, wound healing, and biofilm inhibition/eradication are also known for some of them [20,21,22]. By contrast to conventional antibiotics, antibacterial cell killing of AMPs largely emanate from the disruption of membrane structures. The so called “membranolytic” activity occurs due to the formation of amphipathic structures while in complex with negatively charged bacterial membranes [23,24,25]. Mechanistic actions of AMPs based on plasma membrane disruption processes are proposed e.g., barrel stave, toroidal pore, and carpets [23,24,25]. Cationic AMPs are often able to permeabilize anionic outer membrane LPS which is a necessary step as a part of the mode of action toward Gram-negative bacteria [26,27,28,29,30]. Concomitantly, AMPs can act as potentiators of conventional antibiotics to kill Gram-negative bacteria [11,31,32,33,34]. Atomic resolution structures of some of the potent AMPs in complex with outer membrane LPS have revealed correlations with anti-bacterial activity [35,36,37,38,39,40]. Importantly, activity of AMPs toward antibiotic resistant bacteria is providing an excellent therapeutic avenue for the further development of antimicrobials [41,42,43,44,45]. Emergence of bacterial resistance to AMPs has been found to be limited presumably due to unfavorable evolutionary pressure to change membrane compositions [46,47]. AMP-derived analogs, AMP-conjugated polymers, and AMP nanoparticles are among the robust candidates as therapeutics against drug-resistant bacteria [48,49,50].

Here, we review atomic resolution structures, mode of action of representative potent AMPs which are or were examined in clinical trials at various stages. The review article is intended to deliver an integrative perspective of the clinically tested AMPs of diverse structures and chemical compositions which we believe can be further exploited for rational development of antimicrobials in the fight against bacterial AMR.

## 2. Polymyxins

Polymyxins are a family of cyclic lipopeptides isolated as early as 1947 from spore-forming Gram-positive bacteria *Bacillus polymyxa* [51]. Polymyxins are produced from non-ribosomal peptide synthetase system (NRPS) employing enzymes PmxA, PmxB, and PmxC [52]. The chemical structures of polymyxins, polymyxin B (B1, B2), and polymyxin E (colistin), vary slightly from each other (Figure 1A, Table 1). The cyclic structures of both polymyxin B (PMB) and polymyxin E (PME) contain four cationic diamino-butyric acid (Dab) at positions 4, 5, 8, and 9 and a polar residue Thr 10 [53]. PMB contains an aromatic residue D-Phe6, whereas the 6th position in PME is substituted with residue D-Leu.

The linear tri-peptide segment of PMB and PME consists of residues Dab1-Thr2-Dab3. The sidechain amine group (NH_2_) of residue Dab4 forms a peptide-like covalent bond with the carboxylate group of residue Thr10 conferring 23-atoms heptacyclic ring. Residue Dab1 of PMB and PME is acylated with fatty acids either methyloctanoyl (MeC8) or methylhaptanoyl (MeC7) named as variants PMB1/PME1 or PMB2/PME2, respectively. Polymyxins are highly effective antibiotic peptides against many Gram-negative strains including *E. coli*, *P. aeruginosa*, *K. pneumoniae*, *A.*
*baumannii*. However, clinical use of polymyxins was abandoned in sixties due to observed nephrotoxicity upon discovery of safer new antibiotics [55]. In recent years, despite the known toxicity, polymyxins are reintroduced in clinical usage as a last resort drug option for the hard-to-treat infections of multi-antibiotic-resistant Gram-negative bacteria [51,56,57,58,59]. Optimal dosages and guidelines are recommended for the administration of polymyxins that may potentially reduce acute kidney injury [56,57,58,59]. In this respect, PMB could be considered as a suitable therapeutic option in comparison to PME or colistin methanesulphonate. As PMB demonstrates a lower level of nephrotoxicity and can be retained at a higher plasma level required of the treatment of pathogenic infections [60]. However, non-toxic analogs of PMB with high anti-bacterial activity are essential and some analogs of PMB were tested in clinical trials (Table 1). Among these derivatives, NAB739 and NAB815, developed by Vaara and Northern Antibiotics are well characterized and were found to be less nephrotoxic with the ability to kill drug-resistant Gram-negative bacteria [51,58,61]. Notably, in these analogs two cationic Dab residues, Dab1 and Dab3, were either substituted or deleted achieving lowered toxicity. SPR206, in phase I clinical trial by Spero therapeutics, a more potent analog, contains a shorter linear part and a different composition of fatty acyl chain [51,58,61]. SPR741 a unique derivative of PMB which lacks much of the anti-bacterial activity, however, acts as a strong antibiotics potentiator. Design of SPR741 is akin to PMB nonapeptide (PMBN) which is known to sensitize outer membrane LPS of Gram-negative bacteria [58,59]. SPR741 has passed phase I clinical trial and now in further development.

Recurrent infections of MDR Gram-negative bacteria have re-introduced direct use of nephrotoxic polymyxins and further development of non-toxic PMB analogs. However, mode of action of polymyxins and their derived analogs are yet to be fully understood. Binding of polymyxins to lipopolysaccharide (LPS) in disruption of the outer membrane perhaps is the leading mechanism of bacterial cell death [52,62,63,64]. Other secondary mechanisms are reported including ribosomal RNA binding and oxidative stress/free radical generation [64]. Atomic resolution structures of PMBN, PMB, PME were determined in LPS micelle by use of NMR spectroscopy (Figure 1B) [35,65,66,67]. The docked structure of PMB-LPS complex delineated that cyclic region of PMB binds to lipid A moiety of LPS predominantly by ionic/salt bridge interactions (Figure 1C). In particular, negatively charged bis-phosphate groups of the lipid A establish multiple salt bridges and hydrogen bonds with the sidechains of cationic Dab residues in the hepta-residue ring of PMB [35,65,66,67]. The long fatty acyl chains of LPS/lipid A might have complementary packing interactions with the N-terminal octanoyl acyl chain of PMB. Analogs lacking either N-terminal acyl chain or cationic Dab residues in the cyclic domain of PMB are deduced to be of limited anti-bacterial activity, suggesting validity of the structural models. Direct interactions between PMB and LPS were mapped by STD-NMR studies at atomic resolution (Figure 1D) [54]. The aromatic and non-polar sidechains of residues D-Phe5 and Leu6 were observed to be in proximity with LPS micelle. Moreover, the methyl group and a part of the acyl chain of methyl octanate established close interactions with LPS. By contrast, the cyclic backbone of PMB demonstrated only limited STD effect suggesting lesser contact with LPS micelle. 3-D structures and LPS interactions of PMB at atomic-resolution provided important mechanistic insights in outer membrane recognition and neutralization of endotoxin. However, the exact mechanism of bacterial cell death or lysis of polymyxins are yet to be completely determined. It remains unclear how PMB permeabilizes inner or plasma membrane of bacteria, whether intracellular targets interactions are necessary for cell death. Studying PMB in recently developed model membrane systems e.g., bicelle, lipid nano-disk could be useful for further understanding of mode of action.

## 3. β-Sheet AMPs: Protegrins and Thanatin

β-sheet or β-hairpin AMPs are well-folded even in the absence of bacterial membrane, whereas helical AMPs often lack folded conformations in free solution. The stable β-sheet structures are often supported by inter-strand disulfide bond(s). Protegrins (PGs) constitute 16–18-residue long Arg rich AMPs identified from leucocyte of pig (Figure 2). The β-hairpin structures in PGs are maintained by two antiparallel β-strands with disulfide bonds between residues Cys6-Cys15 and Cys8-Cys13. Broad spectrum antibacterial activity under physiological salt solutions had drawn considerable attention for therapeutic developments of PGs [68,69]. Where PG-1 represents the well-studied member of the group. PG-1 displayed low minimal inhibitory concentration (MIC), ranging between 0.5 to 5 μM, against several strains of Gram-negative and Gram-positive bacteria [68,69,70,71]. Iseganan or IB367 is a synthetic variant of PG-1 developed by IntraBiotics Pharmaceuticals through extensive structure activity studies or SAR [72]. IB367 was subjected to several phases of clinical trials targeting number of conditions e.g., oral mucositis in cancer patients, prevention of ventilator associated pneumonia, cystic fibrosis and mycoses [73,74]. However, the status of these clinical trials of iseganan at present is unknown.

Mechanism of antimicrobial activity of PG-1 has been investigated by several laboratories [75,76,77]. PG-1 causes cell lysis by disrupting bacterial membranes with no known intra-cellular targets [75], as the D enantiomeric form of PG-1 is equally active [75]. PG-1 preferentially interacts with negatively charged membrane lipids over neutral lipids. PG-1 can be readily inserted into lipid films and lipid monolayers consisted of LPS/lipid A and phosphatidyl glycerol. Therefore, PG-1 has been hypothesized to kill bacterial cells by permeabilizing both the outer and inner membranes [75,76]. The high cationicity of PG-1 from six Arg residues is necessary for the broad-spectrum antibacterial activity. Since IB484, a PG-1 analog containing three Arg residues, is largely inactive against Gram-negative *P. aeruginosa*, it can however be toxic to Gram-positive *S. aureus* strain [78]. Structural studies of PG-1 were investigated in solution, membrane mimic micelles, and membrane bilayers. 

Free PG-1 folds into a canonical monomeric β-hairpin structure whereas oligomeric forms could be detected in solution containing zwitterionic detergent micelle (Figure 3A) [77,79]. Solid state NMR revealed that PG-1 may form pore-like structure in model membrane that is rich in negatively charged lipids [76] (Figure 3B). The water filled pore structure is maintained by oligomeric PG-1 and reorganized lipid chains. Interestingly, PG-1 was unable to induce similar pore structure in cholesterol-rich zwitterionic lipid bilayer mimicking eukaryotic membrane compositions (Figure 3D). The structural study indicated mode of action of PG-1 in bacterial membrane lysis. Further, solid state NMR studies demonstrated that PG-1 may form barrel-stave-like pore opposed to toroidal pore in the inner membrane lipid bilayer [80]. Cys-deleted PG-1 (CDP) was investigated for antimicrobial activity, LPS outer membrane binding and structures (Figure 2). CDP inhibited growth of several strains of Gram-negative and Gram-positive bacteria and assumed β-hairpin structure in complex with LPS micelle (Figure 3C). Mode of action of PG-1 and clinical trials of PG-1 derived analogs clearly demonstrated that β-sheet AMPs provide vital lead for the development of therapeutics against MDR pathogens [81].

Next, we discuss thanatin a single disulfide bonded β-sheet cationic AMP demonstrating potent broad-spectrum activity against bacteria (Gram-negative and positive) and fungi [82,83]. The 21-amino-acid-long thanatin peptide (GSKKPVPIIYCNRRTGKCQRM-amide) contains a disulfide bond between Cys11 and Cys 18, with a pI of 10.48. Thanatin has very low hemolytic activity and low toxicity to human cells [82,84]. Thanatin was first isolated and characterized from hemolymph of insect *Podisus maculiventris* (spined soldier bug) following bacterial infection [82]. Preclinical studies in model animals showed high doses of thanatin is well tolerated and could significantly lower bacterial loads of Gram-negative MDR *E. coli* strains [85,86]. Native thanatin and its derived analogs hold a high promise for advanced stages of clinical trials for the potential treatment of MDR Gram-negative infections. Moreover, recent studies showed thanatin utilizes a unique mode of action in killing Gram-negative bacteria. Thanatin efficiently binds to lipopolysaccharide (LPS) at the outer membrane of Gram-negative bacteria yielding a highly permeabilized or disrupted outer membrane [86,87]. The critical feature of mode of action of thanatin is revealed from high affinity binding to LPS transport protein complexes LptA and LptD [88,89,90]. An essential seven member Lpt (A-G) protein complex is involved in transport and assembly of newly synthesized LPS molecules from the inner member to the outer membrane [88,89,90]. Oligomeric LptA is postulated to form a bridge between LptC and LptD, necessary for inter-membrane LPS transport. Binding of thanatin to LptA disrupts the complex with LptC that may inhibit transport of LPS to the outer membrane [88,89,90].

Atomic resolution structures of thanatin are available as a complex of LPS micelle and in complex with LptA_m_, a monomeric shorter variant of LptA [87,89]. In LPS outer membrane, thanatin adopts a dimeric structure whereby residues of the N-terminal β-strand of each sub-unit deduced to be engaged in interfacial interactions (Figure 4A–C). Note that thanatin forms monomeric β-hairpin structures in free solution and also in zwitterionic DPC micelle [87,91]. Therefore, the dimeric structure of thanatin can be considered as a specific consequence of binding with negatively charged LPS outer membrane (Figure 4D). A recent study employing analogs of thanatin disclosed an intrinsic propensity toward dimeric structure in LPS outer membrane [92]. By contrast, thanatin/LptA_m_ structure reveals 1:1 complex of monomeric protein and peptide (Figure 4E,F). The N-terminal β-strand, residues P7-N12, of thanatin is in close contact with the N-terminal β-strand of jellyroll β-structure of LptA_m_ (Figure 4F). The parallel arrangement of the two β-strands is stabilized by means of non-polar packing of sidechains of residues, I8, Y10, and partly sidechain of M21, of thanatin with a set of non-polar/aromatic residues of LptA_m_ (Figure 4F). In addition, residues N12, R13, and R14 of thanatin may form polar interactions with LptA_m_ residues (Figure 4F). It is noteworthy that the N-terminal β-strand of thanatin is commonly involved in maintaining interfacial contacts as in dimeric thanatin and also in complex with protein LptA_m_. A recent study demonstrated that the binding affinity of thanatin to LptA_m_ can be correlated with antibacterial activity [92]. Replacement of residue M21 to aromatic residue Phe, or analog thanatinM21F, has a higher binding affinity to LptA_m_ with concomitant increase in antibacterial activity [92]. The Ala substituted analogs, thanatinY10M21 and thantinR13R14, demonstrated reduced binding affinity and a significant loss of antibacterial activity [92]. Polyphor (now Spexis) pharmaceutical is currently developing thanatin or thanatin-based AMPs for hard to treat infections caused by MDR Gram-negative bacteria. Thanatin is an excellent AMP that binds tightly both to LPS and LPS transport protein machinery for efficient bacterial cell killing. At this moment, little SAR studies are known for thanatin which are very much needed to develop novel antibiotics.

## 4. Outer Membrane Protein Targeting Antibiotics (OMPTA)

OMPTA defines a new class of peptides or peptide mimetics which may be highly effective in killing MDR Gram-negative bacteria [93,94,95]. As a mode of action, OPMTA binds both to LPS and outer membrane proteins resulting in specific Gram-negative activity. In one of these endeavors, protegin-1 was utilized as a starting template generating series of peptidomimetics from designed libraries. In particular, 14-residue long backbone cyclized β-hairpin peptides were developed with inclusion of conserved L-Pro1 and D-Pro14 dipeptide motif [96]. Active peptides were screened from these libraries which specifically inhibited only strains of *P. aeruginosa* with MIC value as low as 0.008 μg/mL [96]. One of the candidate peptides termed murepavadin was tested in clinical trials for potential treatment for pneumonia [97]. Murepavadin binds to outer-membrane β-barrel protein LptD one of the components of LPS transport machinery [96]. However, the exact site of binding of murepavadin or any other related peptides to LptD is not known. As such, LptD has a large, conserved C-terminal domain embedded in LPS outer membrane and a relatively short variable N-terminal at the periplasmic domain [98]. The specific *P. aeruginosa* killing activity of murepavadin over other Gram-negative bacteria may be conferred by binding with the periplasmic domain of LptD. Recently, the phase II clinical trial of murepavadin was halted due to occurrence of acute kidney disease and further preclinical development of the peptide has been undertaken [99].

Darobactin, a seven residue (WNWSKSF) peptide, was first isolated from nematode symbiont bacteria *Photorhabdus khanii* HGB1456. Non-ribosomally synthesized darobactin contains usual sidechain-sidechain covalent crosslinking between indole ring of W1 and β-carbon of W3, β-carbon of K5 and indole ring of W3 [100]. Darobactin can kill several Gram-negative bacteria strains in infection animal model but was not effective to tested Gram-positive bacteria. As a mode of action, darobactin directly binds to the outer-membrane protein BamA which is the central component of BamABCDE complex. Binding of darobactin to BamA inhibited chaperon function of the outer membrane protein causing bacterial cell death. BamA/darobactin complex revealed β-sheet like binding of the peptide with the β1-strand of the β-barrel structure of BamA [100,101]. An intriguing peptidomimetic design principle was developed combining structural elements of murepavadin and PMB. To achieve higher outer membrane perturbation and outer membrane protein binding, chimeric peptides were designed with cyclic LPS binding domain of PMB and cyclic β-hairpin of murepavadin [42]. These chimeric peptides exerted a wider spectrum of activity inhibiting growth of several Gram-negative bacteria including PMB resistant strains. As a mode of action, chimeric peptides efficiently permeabilized both outer and inner membranes. In addition, these peptides bind well with the BamA at the outer membrane [42].

## 5. MSI Peptides

The first AMP isolated from the anuran family is magainin which was discovered from the skin of a female African clawed frog (*Xenopus laevis*) by Michael Zasloff in 1987 [102,103]. Members of the magainin family (magainin-1, magainin-2, and PGLa) are cationic peptides which do not assume any preformed secondary structure in free solution but adopt amphipathic α-helical conformations in membranous environments [104,105]. Magainins are non-hemolytic and non-cytotoxic host defense peptides that however display potent activity against broad range of bacteria, fungi, and protozoa [14,103]. Magainin family is made up of two closely similar peptides (magainin-1 and magainin-2), each of which has 23 amino acids and differs by two substitutions at 10th and 22nd position of its primary structure (Figure 5A) [102,103]. It was also found that magainin-2 has comparatively higher activity than magainin-1. Following this discovery, numerous works were done using magainin-2 and its derivatives to understand activity, structure, and mechanism of action. Through extensive SAR analysis, Zasloff and colleagues at Magainin Pharmaceuticals (now Genaera Corporation) synthesized a 22-residue-long cationic MSI-78 peptide that demonstrated higher potency and greater selectivity to microbial cells compared to human red blood cells [104,106]. MSI-78 exhibited a broad spectrum of potent antimicrobial activities against both Gram-positive and Gram-negative bacteria including the pathogens associated with the diabetic foot infections (DFI) [106]. The MIC_50_ and MIC_90_ against all organisms tested from DFI were 16 and 32 μg/mL, respectively [107]. Although MSI-78 peptide (known as pexiganan or Locilex) showed promising outcomes against several in vitro, in vivo, and pre-clinical studies, the Food and Drug Administration (FDA) of USA finally denied the new drug application (NDA) for its topical administration as a result of a deficit over a traditional antibacterial medication [108]. In 2014, Dipexium Pharmaceuticals initiated another phase III clinical trial using pexiganan and ofloxacin (an FDA approved fluoroquinolone antibiotic) in a comparative clinical study for the same use. However, the trial reported a failure in 2017 since the peptide showed unsatisfactory results at sub-inhibitory concentrations against the tested organisms [109]. It is noteworthy to mention that the Locilex (pexiganan topical cream 0.8%) received the advisory from European Medicines Agency (EMA) in 2015 for human and clinical use against DFI.

Extensive biophysical studies of magainins and MSI peptides have established toroidal pore formation mechanism in membrane for bacterial cell killing [104,105]. Notably, MD simulations revealed toroidal pores could be more dynamic or disordered whereby only few peptides may be present at the center of the pore [110]. In terms of the pore size, magainin was found to induce small toroidal pores of ∼2–3 nm in diameter that only allow water and small intracellular molecules to leak out [111,112]. The excellent attributes of the magainin peptides lead to draw extensive attention in understanding the structure and the mode of action exerted by other AMPs. A high-resolution structure of MSI-78 peptide indicated that it can readily form a dimeric antiparallel α -helical coiled-coil like structure in solution of DPC micelle or lipid bicelle [104]. Such an antiparallel structure is stabilized by a “phenylalanine zipper” motif composed of three phenylalanine residues in the peptide [113,114,115]. While MSI-594 with only two phenylalanine residues assumes a monomeric helical structure under similar membrane mimic environment [116]. Several solid-state NMR analyses of MSI-78 and magainin 2 peptides in aligned bilayer or multilamellar vesicles have confirmed that both peptides retained high propensity toward self-association forming a coiled-coil dimer, but MSI-78 can be dimerized at lower concentrations [117,118,119]. The two-dimensional PISEMA (polarization inversion spin exchange at the magic angle) and ^15^N solid-state NMR analysis revealed that peptide may disintegrate bacterial membranes following a toroidal pore-like mechanism rather than barrel stave fashion [117,120]. Differential scanning calorimetry (DSC) along with ^31^P solid-state NMR spectroscopic investigations have further lend credence to the mechanism of action [118]. DSC experiments showed a concentration-dependent increase in the fluid lamellar to the inverted hexagonal phase transition of the bilayer, indicating that pexiganan induces positive curvature strain on the lipid bilayer. According to ^31^P NMR studies, pexiganan also hindered the fluid lamellar to an inverted hexagonal phase transition [118].

MSI-594 is a synthetic hybrid peptide derived from MSI-78 and melittin, originally designed by Genaera Corporation with a view of increased clinical potency and attributes than its parent peptides [123]. Ramamoorthy and coworkers had performed an extensive study comparing structure and activity of the MSI-78 and MSI-594 by applying an array of biophysical techniques [104,117]. Antimicrobial assays showed that both peptides have similar potency against Gram-positive and Gram-negative bacteria [117,123]. CD analysis in the presence of PC/PG lipid vesicles revealed that MSI peptides reorient from random coil to folded conformations, whereas helical content of MSI-594 has been estimated to be higher in comparison to MSI-78 [104,117]. Based on ^15^N solid-state NMR data, these peptides were aligned roughly parallel to the bilayer surface, indicating that they do not work via the barrel-stave membrane-disruption mechanism. DSC and ^31^P NMR studies, on the other hand, delineated that the integration of peptides into lipid bilayers causes positive curvature strain and disorder in the headgroup and acyl chain regions of lipids [104,117,124]. Furthermore, ^31^P NMR findings indicated that the peptide-induced disorder may be affected by lipid compositions of the bilayers. These observations supported a carpet mechanism of membrane disruption for MSI-594, whereas MSI-78 works via a toroidal-type mechanism in POPG bilayers and a carpet mechanism in POPC bilayers, respectively [104,117,123].

In this regard, our group used high-resolution NMR spectroscopic methods in LPS micelles to better understand the interaction of MSI-594 with Gram-negative bacteria [122]. Interestingly, MSI-594 forms a helical hairpin or helix-loop-helix structure in LPS micelles, opposed to the straight helix determined in detergent DPC micelle [122]. The helical hairpin fold of the peptide is sustained by packing interactions among several hydrophobic residues including Phe5, Ile2, Ala9, Ile13, Leu17, and Leu20, and appeared to be responsible for bacterial outer membrane permeabilization [122]. Further, LPS-MSI-594 interactions are detected by aromatic ring of Phe5 and the side-chain methyl groups of Ile2, Ala9, Ile13, Leu17, and Leu20 residues. The functional importance of Phe5 residue stabilizing the hydrophobic hub of the helical hairpin structure was further illustrated by designing an Ala mutant analog peptide (Figure 5B,C). The MSI594F5A mutant showed reduced antibacterial and biophysical activities along with a distinct structural change from helical hairpin to open helical conformation [121]. Studies demonstrated that MSI-594 interacted with the lipid tails in the hydrophobic core of the bilayer leading to the disruption of the membrane integrity [121]. Importantly, investigations based on these hypotheses have revealed that MSI-594 uses a detergent-type mechanism for membrane disintegration causing bacterial cell death [121,122]. Further works pertinent to MSI-594 structures were done via accelerated molecular dynamics (AMD) simulations using eight different bilayer systems (100% POPC or POPG or POPE or POPS and, 7:3 POPC/POPG or POPC/POPS and, 3:1 POPG/POPE or POPE/POPG) [125]. Mammalian zwitterionic bilayer membrane system was mimicked via POPC (outer leaflet) and POPC/POPS (inner leaflet) lipids. Gram-negative outer and inner membrane mimicking bilayers were formed by 7:3 POPC/POPG and 3:1 POPE/POPG lipid ratio, respectively, whereas Gram-positive bacterial membrane was mimicked via 3:1 POPG/POPE lipid ratio [125]. MD analyses demonstrated that MSI-594 peptide can cause considerable disturbances to the homogeneous zwitterionic POPC bilayer system and least disruptions in a neutral POPE bilayer system [125]. In additions, anionic POPG produces the most fluctuations at lower concentrations, such as in 3:1 POPE/POPG and 7:3 POPC/POPG bilayers [125]. Taken together, the detailed MD study depicted an entire scenario of specific membrane interactions of MSI-594 peptide [125].

## 6. LL-37

LL-37 is a cationic α-helical host defense peptide of human that belongs to cathelicidin antimicrobial peptide (CAMP) family [126,127]. LL-37 is produced by many types of epithelial cells, as well as leukocytes such as monocytes, T cells, B cells, and NK cells and mostly stored in the lysosomes of macrophages and polymorphonuclear leukocytes (PMNs) [127]. All cathelicidin AMPs are synthesized as preproproteins (18 KDa) containing a highly conserved N-terminal domain (13.5 KDa) and a vastly diverse antimicrobial domain at the C-terminus (4.5 KDa) [127]. The N-terminal domain is typically 94–114 amino acids long and shares sequence homology with cathelin, a cysteine protease inhibitor identified in swine neutrophils. The C-terminus of human cathelicidin comprises 37-residue long LL-37 peptide (Figure 6A) [128]. The mature, functional form of the peptide is released after proteolytic cleavage of the signal and cathelin domains [128,129]. In 1995, three scientific groups discovered the LL-37 peptide based on the study of highly conserved “cathelin” domain [127,130]. Beside the cell proliferation, immunomodulation, and other signaling roles, the cationic LL-37 peptide (LLGDFFRKSKEKIGKEFKRIVQRIKDFLRNLVPRTES) shows excellent activity against a broad range of Gram-positive and Gram-negative pathogens [131,132]. Since the LL-37 peptide possess several promising therapeutic potentials including the induction of angiogenesis, the ProMore Pharma in Poland completed the phase IIb clinical trial of LL-37 for treatment of venous leg ulcers [132,133].

The multimodal activities of LL-37 have attracted several research groups to study SAR of this peptide [127,132,137]. Amphipathic LL-37 peptide not only targets LPS or LTA but also interacted with negatively charged DNA, RNA, polyribosomes, and bacterial inner membranes containing phosphatidylglycerol lipids [138,139]. Interestingly, multiplicity of interactions is driven by either monomeric, dimeric, tetrameric, or fibrillar structures adopted by LL-37 or its truncated derivatives [140]. 3-D structures of monomeric LL-37 peptide were determined by using NMR in negatively charged (SDS and D8PG) and zwitterionic (DPC) detergent micelles [134,141]. The peptide adopted an amphipathic kinked α-helical structure in all the environments with flexible N- and C-termini but the bent of the helix was significantly different between the two micelles (Figure 6B) [134,141]. In SDS micelle, the kink is generated from Gly14-Glu16 residues whereas in DPC micelle a single residue Lys12 involves the kink conformation. In-depth analysis of the micelle bound structures revealed that the four Phe residues (Phe5, Phe6, Phe17, and Phe27) directly interact with the hydrophobic core of the micelle [134,141]. Recent x-ray crystallographic structure of LL-37 in DPC micelle with 70% 2-methyl-2,4-pentanediol (MPD) showed a monomeric straight α-helical conformation without any curvature (Figure 6C,D) [135]. The overlay of NMR and x-ray structures showed a significant deviation at the N-terminus (L1-K12) but could align well at the C-terminus (Ile13-Ser37) [140]. The x-ray structure also displayed that the peptide becomes stabilized via several salt-bridge/hydrogen bond interactions between residues Asp4/Arg7, Glu16/Arg19, Gln22/Asp26, Asp26/Arg29 in the helical structure [135].

Several biophysical, structural, and sequence-specific truncations-mutation approaches highlighted the functional role of each residue and segment in the LL-37 sequence [140,142,143]. For instance, truncation analysis has indicated that the first four residues (Leu1-Asp4) are not essential for antibacterial activity but are required for peptide oligomerization [142,143]. This notion is supported by many other analogs, including just the LL-37 core region (FK13, Phe17-Arg29), which remained active similarly like LL-37 against pathogens and also retained the anti-tumor activity (Figure 6G) [142,143]. The solution NMR structure of FK-13 in SDS micelle revealed an α-helical structure with partially disordered termini [136]. The N- and C-terminal fragments of the peptide also adopted similar kind of structures in anionic detergent micelles (Figure 6E,F). Removal of Phe at position 17 resulted in the production of KR-12, which also retained antibacterial effectiveness comparable to LL-37 and FK-13 against *Escherichia coli* but no toxicity to host cells [136,144]. Anti-*candida* and anti-*Staphylococcal* activities of KR-12 and KE-18 analogs were recently discovered [145]. KE18, in particular, displayed anti-biofilm efficacy against yeast and bacteria at sub-inhibitory concentrations [145]. More KR-12 variations were investigated, with the less cationic analogues (a5 and a6) demonstrating significant immunomodulatory, antibiofilm, antibacterial, and osteogenic activities [144]. Through replacements of Ser to Ala (LL23A9) and Ser to Val (LL23V9) at the 9th position, LL-23 variants corresponding to 23 N-terminal residues of LL-37 were created [146]. When compared with native LL-37, LL-23V9 peptide showed higher antibacterial and immunosuppressive activity [146]. Another group recently revealed that FK-16 might be used to repurpose traditional medicines like vancomycin to combat antimicrobial resistance [147]. For the neutralization of lipopolysaccharide (LPS) and lipoteichoic acid (LTA), Nell et al. created a series of short peptides based on LL-37 sequence by substituting sequential, charged, and hydrophobic residues. When compared to LL-37, P60.4, a 24-residue derivative, was shown to have similar LPS/LTA neutralization and antibacterial properties, but with low in vivo toxicity toward the audible canal, skin, and eyes [148]. This peptide was subsequently termed OP-145 and was proven to be safe and efficacious in the treatment of chronic otitis media. OP-145 (IGKEFKRIVERIKRFLRELVRPLR) peptide completed phase II clinical trials successfully in 2019 which was sponsored by OctoPlus pharmaceutical company, a subsidiary of Dr. Reddy’s Research, for the treatment of chronic middle ear infections [133,149]. OP-145 showed excellent activity against MRSA strain and was able to reduce growth and biofilm formation of clinically isolated drug-resistant strains in an in vitro set up [149]. However, OP-145 activity has recently been found to be diminished in human plasma. Following that, synthetic antimicrobial and antibiofilm peptides (SAAPs) including SAAP-145, 148, and 276 were developed, which exhibited substantial anti-biofilm activity against a variety of MDR pathogens [41]. Our group also showed an efficient drug delivery system of EFK17a (EFKRIVQRIKDFLRNLV) and its D-amino acid containing derivative EFK17da (E(dF)KR(dI)VQR(dI)KD(dF)LRNLV) via poly(ethyl acrylate-co-methacrylic acid) microgels [150]. Peptide EFK17 was designed based on the core peptide sequence of LL-37. Solution NMR studies and high-resolution structures of the microgel entrapped peptides delineated a folded amphiphilic α-helical conformation of EFK17 peptide while EFK17da assumed a folded conformation, stabilized by a hydrophobic packings consisting of aromatic/aromatic and aliphatic/aromatic interactions [150]. Both peptides showed a broad range of activity against Gram-positive and Gram-negative bacteria while maintaining a low cytotoxic and hemolytic profile [150]. It is noteworthy to mention that the chicken cathelicidin peptides or fowlicidin-1, 2, and 3 contain highly potent antibacterial and LPS neutralizing activities [151]. Antibacterial and LPS neutralizing activities of fowlicidins and its fragments were thoroughly studied in complex with LPS. Solution NMR studies revealed that the two nontoxic active fragments, residues 1–16 or RG16 and residues 8–26 or LK19, of fowlicidin-1 change their conformations from random coil to amphipathic helical structure while in complex with LPS micelle [151]. The LK19 peptide adopted a well-defined α-helical structure with a bend at the middle. On the contrary, the first seven amino acids of RG16 are found to be flexible followed by a helical conformation for the residues L8-A15 [151]. These extensive biophysical and structural studies opened a new arena to design novel nontoxic endotoxin neutralizing molecules.

## 7. De Novo Designed Synthetic Peptides

De novo designed synthetic AMPs can be potentially developed into new class of antibiotics. They are rationally created with an antibacterial pharmacophore while providing chemical structural flexibility to modify for desirable features such as enhanced activity, decreased cytotoxicity, and proteolysis [152]. A range of chemical synthesis methods is utilized to generate a variety of antimicrobial peptides and libraries [153]. Some de novo designed AMPs are known to be displaying excellent broad-spectrum activity against drug-resistant pathogens and removal of preformed biofilms [154]. Here the discussion is limited to certain examples where origin of the peptide is not obtained from any naturally occurring AMPs. Novexatin (NP213) peptide, for example, is a new cyclic, water-soluble antifungal peptide that has been developed exclusively for the topical treatment of onychomycosis [155]. NP213 can penetrate human nails successfully. Following topical application to the skin and nails, NP213 revealed an excellent preclinical and clinical safety profile, with no indication of systemic exposure. NP213 acts on the bacterial and fungal membranes and disrupts them via pore formation [155]. The phase IIa clinical trial of Novexatin was successfully completed in 2018 and is currently under development by Taro Pharmaceuticals USA [155]. WLBU2 is another de novo engineered peptide demonstrating high selectivity, a broad spectrum of activity (against bacteria, virus, fungus), LPS neutralization efficacy and anti-biofilm activity [156]. It also showed excellent synergistic activity with conventional antibiotics, Polymyxin B and other AMPs [156,157]. CD analysis indicated that the peptide does not adopt any predefined structure in free solution but adopts an amphipathic α-helix conformation in negatively charged cell membranes or membrane mimics [156,158]. WLBU2 appears to disrupt bacterial cells akin to LL-37 peptide. Phase I clinical trial of WLBU2 is currently in progress [159]. LTX-109 (Lytixar) is a synthetic antimicrobial peptidomimetic developed by Lytix Biopharma (Oslo, Norway) [160,161]. It is active against a broad range of Gram-positive and Gram-negative bacterial species and rapidly kills bacteria via membrane disruption [161]. Notably, LTX-109 is highly stable against protease degradation. The LTX-109 is a tripeptide containing a modified tryptophan residue and is capped at the C-terminal by an ethyl-phenyl group [160,161]. NMR and MD simulation studies with model membranes showed that the LTX-109 peptide first interacts electrostatically with negatively charged phosphate head groups and subsequently buries the hydrophobic elements (Tbt and C-terminal capping) in the acyl core of membrane [160,161]. LTX-109 is now being tested in phase II clinical studies for the topical treatment of infections caused by MDR pathogens [160,161].

Other notable synthetic AMPs, which are under preclinical trials, are HB1345 and Novarifyn. HB1345 is a synthetic short lipopeptide (decanoyl KFKWPW) developed by HelixBioMedix that exhibits a broad spectrum of activity against Gram-positive and Gram-negative bacteria. The designed peptide specifically binds to LTA and LPS and disrupts the microbial cell wall barrier [162,163]. Novarifyn, developed by NovaBiotics Ltd. (Aberdeen, UK), is highly active against MRSA, *Pseudomonas*, *Acinetobacter,* and *Clostridium* spp [164,165]. In line with this discussion, our group developed a rationally designed synthetic AMP, VG16KRKP (VARGWKRKCPLFGKGG), from the dengue virus fusion protein with a broad spectrum of activity against plant and animal pathogens as well as clinical fungal strains [166,167]. This peptide showed good membrane selectivity and is non-hemolytic and non-cytotoxic in nature. The solution-NMR structure of VG16KRKP in presence of LPS showed a folded conformation with a centrally located turn-like structure stabilized by aromatic-aromatic packing interactions with extended N- and C-termini [166,168]. Our work highlighted potential direct application to treat pathogenic bacterial infection in rice and cabbage, caused by *Xanthomonas oryzae* and *Xanthomonas campestris*, respectively [166]. Dimerization of this peptide (VG16KRKP-dimer) via Cys9 residue or lipidation with four-carbon-long acyl chain attached at the N-terminal resulted in an improved antimicrobial efficacy with two–ten-folds drop in MIC values as well as a 10% increase in endotoxin LPS neutralization [169]. Interestingly, conjugation of the peptide with gold nano particle depicted efficiency in killing of MRSA strain [170]. To increase the positive charge and develop more potent AMP, a shorter analogue VG13P (VARGWGRKCPLFG) was combined with a “KNKSR” moiety from Bovine lactoferrin (WR17) peptide at the N- and C-termini. Thus, the KG18 (KNKSR-VARGWGRKCPLFG) and VR18 (VARGWGRKCPLFG-KNKSR) peptides were synthesized which showed more promising activity than the parent peptides [171]. The VR18 peptide was effective against invasive *P. aeruginosa* strain which is the major causal agent of human corneal keratitis [172]. A recent study showed that VR18 can effectively kill *P. aeruginosa* PAO1 and 6294 strains in an in vitro, ex vivo, and in vivo set-up [172]. Extensive biophysical studies indicated that the peptide selectively acts in a membranolytic fashion against negatively charged bacterial membrane or liposomes [172]. The high-resolution NMR structures in presence of bacterial outer and inner membrane mimicking systems revealed that the peptide can adopt a well-folded amphipathic structure that displays a complementary surface for interactions with LPS [172]. These results motivated us to proceed further to evaluate the pre-clinical potential of the VR18 peptide in near future.

LPS structure was utilized in a separate endeavor of de novo designing antimicrobial and antiendotoxic peptides. In initial designs, a 12-residue peptide, YW12 (H_2_N-YVLWKRKRMIFI-COOH), was synthesized and investigated for lipid interactions and 3-D structure in LPS [173]. Peptide YW12 structure was solved in complex of LPS micelle, by NMR spectroscopy. Atomic-resolution structure of the designed peptide revealed a novel amphipathic β-strands-loop fold whereby the four central cationic residues (KRKR) and hydrophobic residues (YVL, IFI) at the N- and C-termini were distinctly spatially separated into two mini domains. The amphipathic β-sheet structure was found to be maintained by long-range packings between residues W4 and M9. Further, YW12 peptide specifically binds to negatively charged lipids and detergents, whereas peptide’s interactions with zwitterionic micelles was found to be limited [173]. In further design, residue M9 of YW12 peptide was substituted with several aromatic and non-polar residues. These newly designed peptides, termed β-boomerang, with an aromatic residue (Y, F, or W) at position 9 displayed robust structures in LPS micelle which was sustained by aromatic/aromatic packing interactions with residue W4. SAR studies demonstrated strong correlations of the β-boomerang structures with antimicrobial and antiendotoxic activity [174,175]. Further, lipidations of β-boomerang peptides yielded better performing analogs of low bacterial MICs and low hemolytic activity [176]. Notably, specific interactions of β-boomerang peptides with LPS were utilized in designing Gram-negative active AMP-peptide hybrids, bacterial photodynamic therapy, sensitive endotoxin sensors, and smart antibiotic delivery systems [174,177,178,179].

## 8. Conclusions and Future Perspective

Quick and powerful actions are urgently needed to combat the continuous uprising of MDR pathogens. AMPs have already proven their potency to be used as an alternative arsenal against a broad range of pathogens including viruses, bacteria, fungus, and protozoa. Despite past failures, in recent years, a steady increase of AMPs and AMP mimicked antimicrobials in clinical trials is highly encouraging. Failures of AMPs in clinical trials could be attributed to less clear efficacy, inappropriate study design, or inferiority over conventional antibiotics. As a result, in future clinical testing of AMPs, practical techniques should be addressed, as we have gained the following lessons: (1) the use of AMPs may go beyond FDA-approved clinical indications; (2) the most effective doses and administration regimen are to be determined to reduce cytotoxicity of AMPs; (3) molecules efficacy can be demonstrated in equivalence or non-inferiority trials with an antibiotic as a comparator; (4) bacterial resistance development should be included as one of the primary outcome parameters in clinical trials of AMPs; (5) bioavailability and effectiveness need to be increased by employing appropriate delivery methods; and (6) clinical trials may include combinations of AMPs with conventional antibiotics or other drug modalities for enhanced antibacterial activity. Taking these lessons into account, a growing number of AMPs may enter the market as multi-functional, powerful, and long-lasting antimicrobial agents against a variety of infectious disorders.

## Figures and Tables

**Figure 1 ijms-23-04558-f001:**
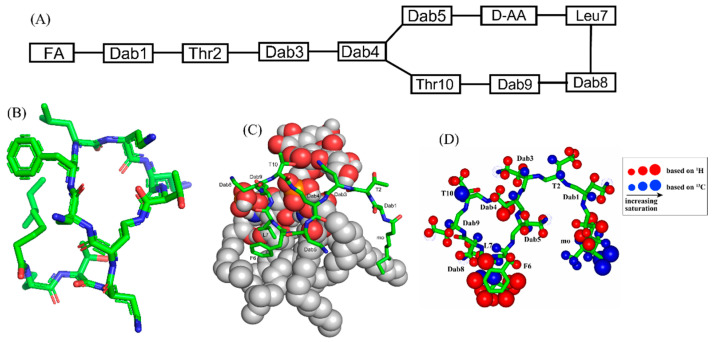
(**A**) The primary structure of PMB. Both PMB and PME contains cationic Dabs at positions 4, 5, 8, and 9 along with a polar Thr10 residue in their cyclic structures. (**B**) 3-D structure of PMB represented by a stick model [35]. (**C**) The docked structure of PMB-LPS complex shows that the cyclic region of PMB (stick model) binds to lipid A moiety of LPS (space filling model) predominantly by ionic/salt bridge interactions. (**D**) Summary of the epitope mapping of PMB in complex of LPS based on ^1^H 1D STD experiments (red color) and natural abundance ^13^C-^1^H HSQC experiments (blue color) [54]. The structure of PMB is represented as solid stick. The protons are depicted as sphere of different sizes based on determined STD values. (**D**) is reproduced from reference 54 upon permission obtained from the publisher.

**Figure 2 ijms-23-04558-f002:**
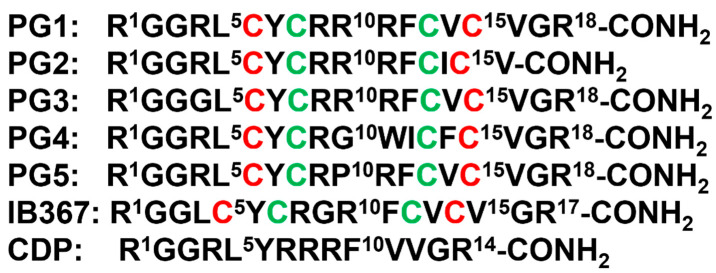
Amino acid sequence of protegrins and its derivatives. The disulfide bonds are formed between Cys6-Cys15 and Cys8-Cys13 residues, depicted by red and green color, respectively.

**Figure 3 ijms-23-04558-f003:**
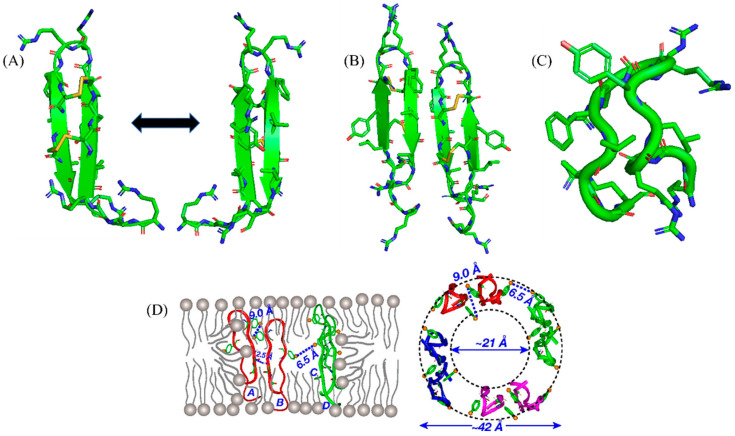
(**A**) In a solution containing zwitterionic detergent micelle, free PG-1 folds into a typical monomeric beta-hairpin structure [77]. (**B**) The solid-state NMR analysis demonstrated that PG-1 may create a pore-like shape in a model membrane rich in negatively charged lipids [78]. (**C**) LPS-bound solution NMR structure of Cys-deleted PG-1 (CDP) [81]. (**D**) The water-filled pore structure is maintained by the reorganization of oligomeric PG-1 and lipid chains. PG-1-like pore shape was not seen in cholesterol-rich zwitterionic lipid bilayers that mimic eukaryotic membrane compositions [76]. (**D**) is reproduced from reference 76 Mani, R.; Cady, S.D.; Tang, M.; Waring, A.J.; Lehrer, R.I.; Hong, M. Membrane-dependent oligomeric structure and pore formation of a beta-hairpin antimicrobial peptide in lipid bilayers from solid-state NMR. *Proc Natl Acad Sci USA*
**2006**, *103*, 16242–16247, doi:10.1073/pnas.0605079103, “Copyright (2006) National Academy of Sciences, USA.”

**Figure 4 ijms-23-04558-f004:**
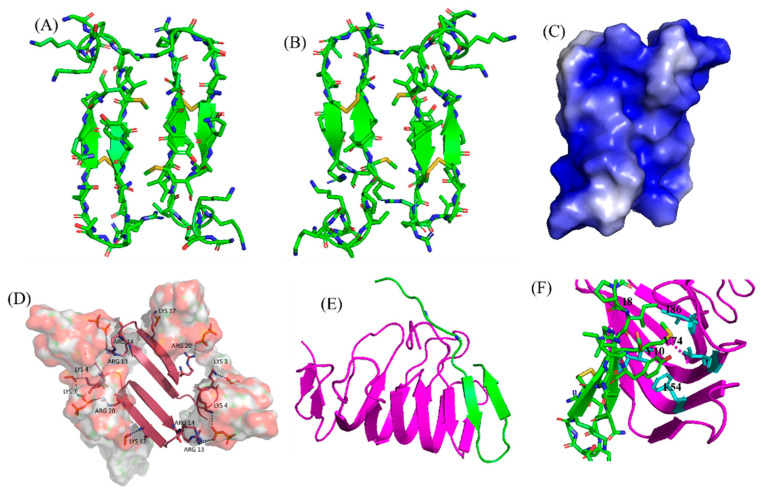
(**A**,**B**) Dimeric β-sheet structure of thanatin as a complex of LPS micelle in two different orientations [87,91]. (**C**) Molecular surface electrostatic potential of the dimeric structure of thanatin in LPS micelles (pdb: 5XO9), colored according to residue type. Blue, cationic and white, non-polar residues [87]. (**D**) Molecular docking of thanatin dimer with LPS highlighting interactions of cationic resides with negatively charged phosphate head groups of LPS [87]. (**E**) NMR derived structure of thanatin/LptA_m_ complex (pdb: 6GD5). Thanatin is in green and LptAm is in purple ribbon. (**F**) The N-terminal beta-strand, residues P7-N12, of thanatin is in proximity with the N-terminal beta-strand of LptA_m_ jellyroll beta-structure. A parallel organization of the two beta-strands is maintained by close packing of thanatin residues I8, Y10, and M21 with a set of non-polar/aromatic LptA_m_ residues [92].

**Figure 5 ijms-23-04558-f005:**
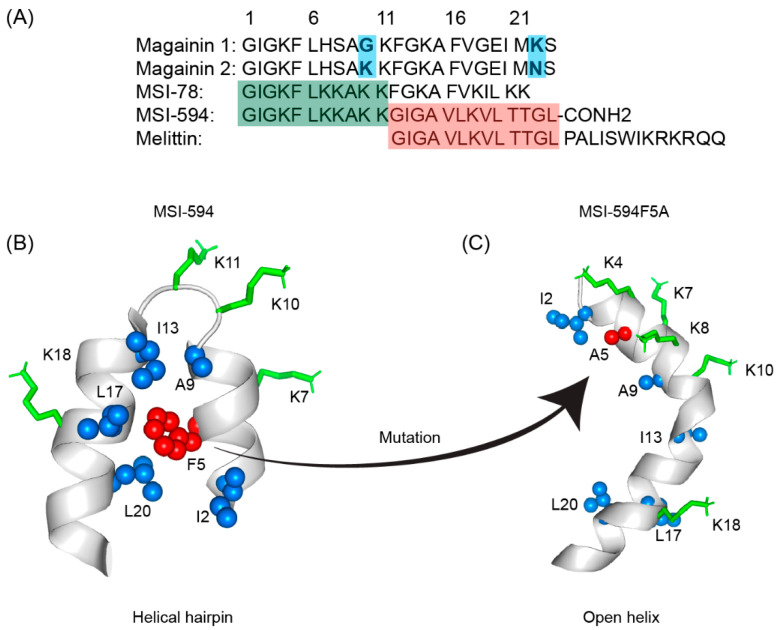
(**A**) Amino acid sequences of magainin 1, magainin 2, MSI-78, MSI-594, and melittin. The sequence of MSI-594 was derived from MSI-78 and melittin, as indicated by the green and maroon boxes, respectively. (**B**) MSI-594 adopted helical hairpin or helix-loop-helix structure in presence of LPS, while (**C**) mutation of Phe5Ala converted the helical hairpin structure to open helix conformation in the same environment [121,122].

**Figure 6 ijms-23-04558-f006:**
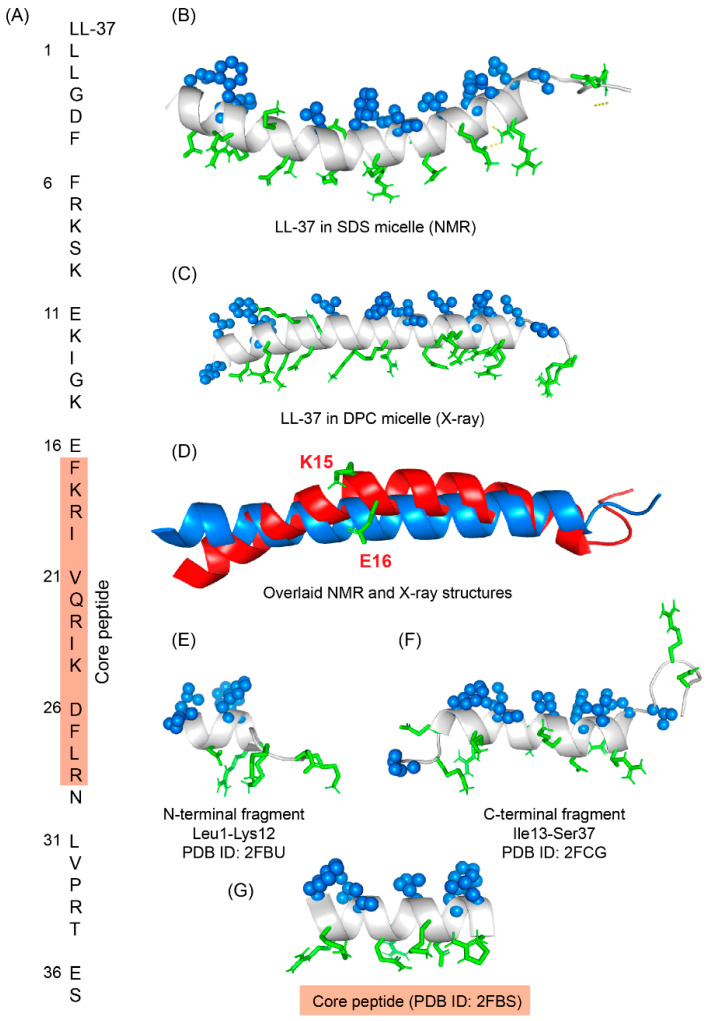
(**A**) Amino acid sequence of LL-37 peptide; core peptide is highlighted by maroon-colored box. (**B**) LL37 adopted a twisted amphipathic α-helical conformation in presence of SDS micelle (PDB: 2K6O) [134]. (**C**) In presence of DPC micelle, the peptide obtained a straight helical conformation as seen in the X-ray crystallographic analysis (PDB: 5NMN) [135]. (**D**) The overlaid pdb of NMR and crystallographic structures showed that the SDS bound LL-37 has a kink at residues Lys15 and Glu16. (**E**–**G**) The N-terminal, C-terminal, and core peptide fragments also obtained an amphipathic conformation in presence of SDS micelle [136].

**Table 1 ijms-23-04558-t001:** Polymyxin B, colistin, and the novel polymyxin based analogs with improved efficacy in animal infection models [51] ^a,b^.

Compound	FA	Sequence
Polymyxin B (PMB)	Methyloctanoyl/methylheptanoyl	Dab1-Thr2-Dab3-cy[Dab4-Dab5-DPhe6-Leu7-Dab8-Dab8-Thr10]
Polymyxin E (PME, Colistin)	Methyloctanoyl/methylheptanoyl	Dab1-Thr2-Dab3-cy[Dab4-Dab5-DLeu6-Leu7-Dab8-Dab8-Thr10]
FADD002	Octanoyl	Dab1-Thr2-Dab3-cy[Dab4-Dab5-DAda6-Leu7-Dab8-Dab8-Thr10]
FADD287	Octanoyl	Dab1-Thr2-Dap3-cy[Dab4-Dab5-DLeu6-Abu7-Dab8-Dab8-Thr10]
CA284	(*S*)-1-(2-methylpropyl)-piperazine-2-carbonyl+	Thr1-Dab2-cy[Dab3-Dab4-DPhe-Leu6-Dab7-Dab8-Thr9]
SPR206	(3*S*)-4-amino-3-(3-chlorophenyl)butanoyl	Thr1-Dab2-cy[Dab3-Dab4-DPhe-Leu6-Dab7-Dab8-Thr9]
MicuRx-12	3-(2,2-dimethyl-butanoyloxy)-propanoyl (ester bond)	Dab1-Thr2-Dab3-cy[Dab4-Dab5-DPhe6-Leu7-Dab8-Dab8-Thr10]
NAB379	Octanoyl	Thr1-DSer2-cy[Dab3-Dab4-DPhe5-Leu6-Dab7-Dab8-Thr9]
NAB815	Octanoyl	Dab1-Thr2-DThr3-cy[Dab4-Dab5-DPhe6-Leu7-Abu8-Dab8-Thr10]

^a^ Amino acyl residues that differ from those in polymyxin B are marked in Red, ^b^ Abu, aminobutyryl; Ada, aminodecanoyl; Dap, diaminopropionyl; cy, cyclic region indicated with brackets; Dab, diaminobutyryl; FA, fatty acyl.

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
