# Peer review of "Atomic-Resolution Structures and Mode of Action of Clinically Relevant Antimicrobial Peptides"

_ijms, 2022, doi:10.3390/ijms23094558_

Round 1

Reviewer 1 Report

Antimicrobial peptides (AMPs) are small aminoacid-based antimicrobials produced by taxonomically diverse organisms. These potent natural antimicrobials are promising templates for drug developement, in particular to mitigate the current and emerging threat of AMR.

This review (according to the abstract) aims to discuss antimicrobial peptides, with particular emphasis on their 3-D structure and their mechanism of action including a good revision of the literature.

The manuscript is generally well-written but the authors are advised to seek the assistance of a native speaker and/or an experienced editor before it can be accepted for publication

However, few minor points need to be addressed before the publication:

Recent literature data report AMPs with intracellular targhet, which describe their mechanism of action. This aspect is absent in the manuscript. I suggest to add this section in the manuscript, to improve the paper.

Line 62 to 71 – the authors repeat “AMP” many times, I suggest removing and/or change the format of the text.

Line 126 – remove comma after As

Line 312 – Change BAMA with BamA.

Line 346 – g/mL? check this concentration please.

Author Response

Reviewer 1

The manuscript is generally well-written but the authors are advised to seek the assistance of a native speaker and/or an experienced editor before it can be accepted for publication.

: We have now thoroughly edited the manuscript with the help of a native speaker.

However, few minor points need to be addressed before the publication:

Recent literature data report AMPs with intracellular targhet, which describe their mechanism of action. This aspect is absent in the manuscript. I suggest to add this section in the manuscript, to improve the paper.

: Please note section 4 OMPTA and thanatin target intracellular proteins.

Line 62 to 71 – the authors repeat “AMP” many times, I suggest removing and/or change the format of the text.

: We have now revised the text from line 62 to 71.

Line 126 – remove comma after As

: Edited

Line 312 – Change BAMA with BamA.

: Edited

Line 346 – g/mL? check this concentration please

: Unit has now been changed to mg/mL

Reviewer 2 Report

In this manuscript, the authors review the current knowledge about some antimicrobial peptides from structure, mode of action to current development and clinical stage progression by pharmaceutical companies. This review is very interesting and provides a large and complete overview of the field with also special focus on the authors own contribution to this research topic.

I have minor comments.

- Line 102 and 163, the numbering for residues in PMB is not clear. D-Phe should be 6 and Leu 7.

- As most abbreviations are indicated, some are missing. In particular CDC (line 38), SAR (line 184)

- Paragraph 3 started by "by contrast to helical AMPs" (line 174) but the helical peptides have not been yet introduced.

- Though ref 100 and ref 101 (line 316) shows that darobactin binds to BamA-β-barrel, ref 100 is by using NMR and ref 101 by cryo-EM

- line 291: "deigned libraries", misspelled for designed?

Author Response

Reviewer 2

- Line 102 and 163, the numbering for residues in PMB is not clear. D-Phe should be 6 and Leu 7.

: We have now corrected the numbering.

- As most abbreviations are indicated, some are missing. In particular CDC (line 38), SAR (line 184)

: We have now provided abbreviation for CDC and SAR.

- Paragraph 3 started by "by contrast to helical AMPs" (line 174) but the helical peptides have not been yet introduced.

: We have now rephrased this sentence.

- Though ref 100 and ref 101 (line 316) shows that darobactin binds to BamA-β-barrel, ref 100 is by using NMR and ref 101 by cryo-EM

: We have corrected the statement

- line 291: "deigned libraries", misspelled for designed?

: Corrected
